# Characterizing the variation in chromosome structure ensembles in the context of the nuclear microenvironment

**Priyojit Das**[1], **Tongye Shen**[1,2], **Rachel Patton McCord**[1,2]\*

1 UT-ORNL Graduate School of Genome Science and Technology, University of Tennessee, Knoxville, Tennessee, United States of America, 2 Biochemistry & Cellular and Molecular Biology, University of Tennessee, Knoxville, Tennessee, United States of America

\* rmccord@utk.edu

**Data Availability Statement:** Example analysis code is provided on the ChromosomeStructuralEnsembleCharacterization GitHub repository (https://github.com/

## Abstract

Inside the nucleus, chromosomes are subjected to direct physical interaction between different components, active forces, and thermal noise, leading to the formation of an ensemble of three-dimensional structures. However, it is still not well understood to what extent and how the structural ensemble varies from one chromosome region or cell-type to another. We designed a statistical analysis technique and applied it to single-cell chromosome imaging data to reveal the heterogeneity of individual chromosome structures. By analyzing the resulting structural landscape, we find that the largest dynamic variation is the overall radius of gyration of the chromatin region, followed by domain reorganization within the region. By comparing different human cell-lines and experimental perturbation data using this statistical analysis technique and a network-based similarity quantification approach, we identify both cell-type and condition-specific features of the structural landscapes. We identify a relationship between epigenetic state and the properties of chromosome structure fluctuation and validate this relationship through polymer simulations. Overall, our study suggests that the types of variation in a chromosome structure ensemble are cell-type as well as region-specific and can be attributed to constraints placed on the structure by factors such as variation in epigenetic state.

## Author summary

Recent work has revealed principles of how chromosomes are folded and structured inside the human nucleus. It is now even possible to microscopically trace the path of chromosomes in 3D in individual cells. With this data, we can start to examine how much variation exists in chromosome structure and what biological factors may restrict or enhance this variation. Are chromosomes stuck in just a few possible positions or do they move around more freely, sampling many configurations? Here, we use a mathematical approach to compare chromosome structure variation in different cell types, at different locations along the genome, and when key structural proteins are removed. Through

rpmccordlab/
ChromosomeStructuralEnsembleCharacterization).

**Funding:** This work was supported by the National Institutes of Health [NIGMS grant R35GM133557 to R.P.M]. The funders had no role in study design, data collection and analysis, decision to publish, or preparation of the manuscript.

**Competing interests:** The authors have declared that no competing interests exist.

these comparisons and dynamic simulations of chromosome behavior, we identify factors that may constrain or promote variation in chromosome structure.

## Introduction

In eukaryotic cells, long genomic DNA is packed within a relatively small nucleus, requiring chromosomes to fold into a multi-scale 3D spatial organization [1]. At the kilobase length scale, loops form between different chromatin regions [2–4]. Some loops are formed by the interaction of cohesin-mediated loop extrusion and boundary factors, a process which can create domains of enriched contacts called Topologically Associated Domains (TADs) at the scale of 100s of kilobases [5–7]. Through processes such as phase separation and tethering to nuclear structures the chromatin is further organized at the megabase scale into separate euchromatin and heterochromatin spatial regions, known as A/B compartments [8–12]. Finally at the scale of the whole nucleus, chromosomes tend to form their own territories [13–15]. Each of these structural features act independently as well as synergistically to contribute to different biological processes, as reported by an ever-increasing number of studies. For example, chromatin loops and TAD boundary positioning can influence enhancer-promoter interactions and gene regulation [16–18]. This multi-layered organization of 3D genome is not static: the chromatin fiber is semiflexible in nature and is subjected to forces generated by ATP-driven molecular machines along with thermal fluctuations which generate chromosome structure heterogeneity on a temporal scale [19–21]. Some chromosome structures (such as the positioning of whole chromosomes or nuclear lamina associations) are stable on a timescale of hours [22,23], but heterogeneity in these structures can arise, for example, due to stochastic variations that occur as each cell rebuilds its chromosome structure after mitosis [23,24]. Overall, therefore, variations observed in a chromosome structure ensemble reflects a combination of both cell-to-cell and temporal structural heterogeneity. Measuring the heterogeneity of the structural ensemble may reveal detailed folding mechanisms and subpopulations of cells with different chromosome structures that mediate different biological outcomes [25].

Several key molecular and imaging techniques to characterize chromosome structure have been developed in recent years [26,27]. Chromosome conformation capture (3C) based techniques, including Hi-C, can map the spatial organization of the whole genome in terms of contacts between genomic regions. However, when Hi-C and its derivatives are applied to millions of cells in bulk, it is challenging to study the heterogeneity of chromosome structures from the resulting population averaged measurements. Several polymer- and restraint- based models have been developed to reconstruct ensembles of structures from such population averaged Hi-C data [28–32]. However, there are usually multiple possible theoretical ensembles that could fit the population average, so without single cell measurements for comparison, it is difficult to know whether models derived from bulk data capture the variation present in true biological systems. To resolve this issue, researchers have started to develop single-cell counterparts of Hi-C and related techniques [33–35]. However, these single cell contact capture techniques often suffer from low coverage, which makes it difficult to characterize the structural heterogeneity within a particular cell-type.

In contrast, oligonucleotide-based FISH imaging techniques have enabled the tracing of chromatin regions at high resolution in thousands of single cells [25,36–40]. These techniques can generate distance distribution maps between imaged points along a chromosome that resemble Hi-C contact maps. By analyzing these single-cell structures at different resolutions, recent studies have started to characterize the heterogeneity associated with the structural

ensemble. For example, it has been revealed that the inactive chrX exhibits not only canonically expected compact structures but also more chromosome domain inter-mingling as compared to the active chrX [36]. Another study revealed that domain-like structure also exists on the inactive chrX at the single-cell level, but such structures are averaged out and not visible in bulk experiments [25]. Bintu and co-authors reported the existence of TAD-like structures and sharp domain boundaries at a single cell level [37]. Though the boundary positions vary from one cell to another, they show higher preference for structural protein binding sites. By analyzing the single-cell whole chromosome tracing data at a higher resolution, researchers identified a substantial number of TAD-like domains in single cells that cross compartment boundaries observed in bulk data [38].

While initial studies focused on showing that variability in chromosome structures exists, recent work has begun to turn toward the question of how and why this variability arises [30,41,42]. For example, transcriptional regulation has been proposed to influence an observed two-state transition between closed and open dumbbell-like conformations of a short chromatin segment spanning a TAD boundary [41]. Other groups have analyzed the chromatin tracing data to evaluate the effects of the structural protein cohesin on the dynamics of TAD-like structures. Another study used a folding coordinate to represent the progression of TAD formation. These analyses showed that in the absence of structural protein cohesin, a local chromatin region may still exhibit TAD-like structures in individual snapshots, but these structures are less stable due to the entropy gain of the unfolded conformations. [42]. This has the effect of converting phase separated globules into more random coil-like state [30].

Although these recent studies have started to shed light on the fundamental mechanisms of local chromatin folding at the single-cell level, it is unclear how the chromosome structure variation differs between cell types or chromosome regions. Here, we focus on quantifying the full structure variation landscape rather than identifying multiple discrete structural states. Support for this view comes from a variety of recent studies. A FISH imaging study in *Caenorhabditis elegans* showed that chrV exhibits four major conformational clusters during early embryonic development, but that the structure ensembles corresponding to different clusters exhibit a broad and continuous distribution rather than a handful of discrete conformational states [39]. Similarly, within a particular cell-type, the single-cell structure ensemble can be described as a distribution of conformations around the bulk average structure (mean). We do not know how this fluctuation around an average structure (variance) varies in a cell-type specific manner. This question becomes particularly interesting when the average structural organization looks similar across different cell types or conditions, but there may be different underlying structural ensembles that lead to similar averages. A logical follow up question is what factors might contribute to the extent of structural variation. There are various essential factors such as epigenetic marks, associated structural proteins, and nuclear lamina associations that vary in a cell-type specific manner and contribute to the organization of 3D genome [43–45]. Therefore, investigating the role of these different factors in the variation of chromosome structural ensemble is of high importance.

To answer these questions about chromosome structural ensembles, we designed a statistical pipeline to analyze structural data obtained from single-cell chromatin tracing. Using this pipeline, we first characterized the landscape of single-cell structures across different cell-types. We then identified cell-type specific differences in the structural ensembles through joint analysis of structural landscapes. To quantify these differences, we designed two different network-based measurements–one to quantify the degree of continuity across the chromosome structural ensemble (ensemble structure similarity index) and another to quantify the degree of structural variation of local genomic regions across the population (bin similarity index). Using these two measurements, we quantified the ensemble variation and local

structural variation across cell-types and chromosome regions at different genomic resolutions. We identify overall locus compaction and domain boundary shifting as two primary but independent modes of structural fluctuation that are observed across all length scales. We observe differences in chromosome ensemble structural similarity between regions that have different epigenetic state profiles, and we are able to demonstrate that interactions governed by epigenetic state can help explain these differences in the type of structure variation exhibited by the chromosome region.

## Materials and methods

The statistical chromosome conformation analysis pipeline used in this work can be broadly divided into two parts. In the first part, the single-cell chromosome imaging data (of different genomic resolutions) is analyzed statistically to compare the structures in a pairwise fashion. Next, a structural similarity measurement is applied to quantify the degree of continuity present in the chromosome structural ensemble. In addition, the pairwise similarity data is used to construct a structural landscape to arrange the single-cell chromosome structures according to their individual and bulk structural properties (Fig 1).

### Construction of distance matrices from single-cell chromosome imaging data

For this work, we have examined experimental single-cell chromosome imaging data at three different resolutions, each of which reflects different levels of structural organization of the 3D genome at different genomic length scales. The single-cell chromosome imaging data obtained from Bintu *et al.* [37]. has the highest genomic resolution of 30 kb. Here, the authors traced the conformations of multiple ~ 2–2.5 Mb spanning chromatin regions of chr21, partitioned into consecutive 30 kb segments in single-cells across different cell-types (e.g., IMR90 –human fetal lung fibroblast cells, K562 –human erythroleukemic cells, A549 –human lung adenocarcinoma cells, and HCT116 –human colon cancer cells) during different physiological and perturbed conditions. The second set of IMR90 chromatin tracing data analyzed here is obtained from Su *et al.* [38], where the single-cell conformations of chr21 non-repetitive region were traced using consecutive 50 kb bins. In addition, that dataset also provides chr2 conformations captured at the same 50 kb resolution, however with an interval of 250 kb between two genomic regions, due to very large size of human chr2. Both the Bintu *et al.* and Su *et al.* data capture chromatin conformations at a comparable resolution (30 and 50 kb respectively), differing only in the size of the genomic region analyzed. Whereas the Bintu *et al.* data primarily represents the structural heterogeneity at the level of specific TADs and sub-TADs, the Su *et al.* data captures the heterogeneity of almost the whole chromosome. Finally, we also analyzed chromosome structural heterogeneity of the whole active and inactive chrX of IMR90, obtained from Wang *et al.* [36]. In this dataset, TADs are imaged as the basic subunits along the chromosome, so the resolution of this data is lower (ranging from 440 kb to 7.88 Mb) and reports on the conformations of compartments and the whole chromosome rather than the within-TAD level.

For all these different datasets, the authors reported the single-cell conformations of the chromosomes in a 3D Cartesian coordinate system where each basic genomic region (e.g., 30 or 50 kb genomic region or TAD; from here termed as bin) has X-, Y-, Z- coordinate values. However, in some single cell structures the 3D positions of some of the bins are missing due to technical difficulties. Therefore, to decrease the effect of missing data, we consider only structures with more than ~ 90% measured bins, and the values for missing bins in those structures are assigned by nearest-neighbor interpolation. The 3D positions of chromosomes are also

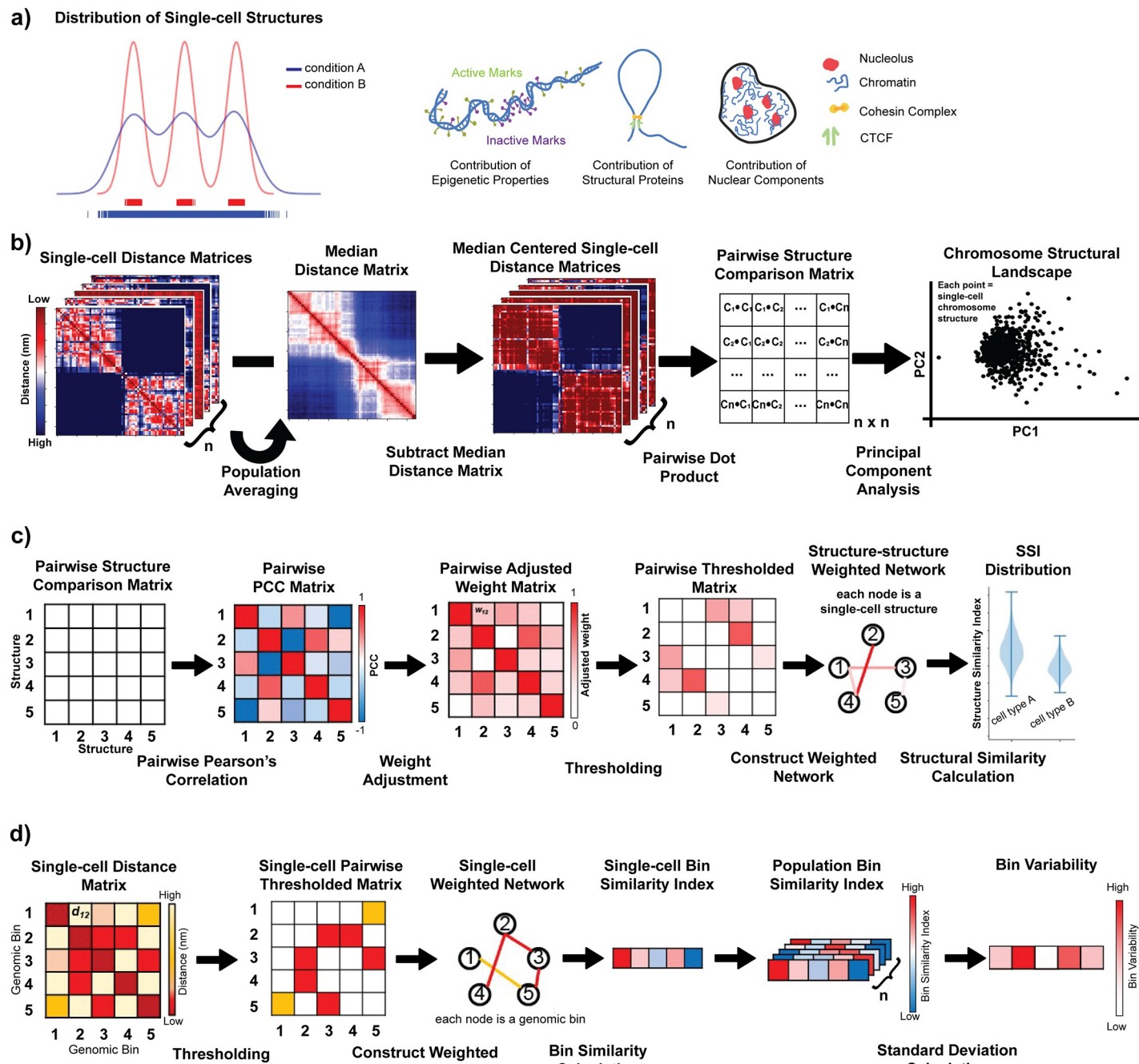

**Fig 1. Schematic representation of the statistical pipeline used to quantify conformational diversity present in single-cell chromosome distance matrices. a)** Cartoon representation of potential differences in chromosome structure landscapes (left). Condition A has a continuous landscape including all structures along a trajectory, while in Condition B, chromosomes fall into 3 major structural clusters. (Right) Conditions and factors that may influence this chromosome structure landscape. **b)** Each pair of median-centered chromosome distance matrices are compared in a pairwise fashion. PCA is then performed on the pairwise structure comparison matrix and transformed data is projected in 2D space using PC1 and PC2 to create the "chromosome structural landscape". **c)** To calculate chromosome ensemble structure similarity index (SSI), we construct a Pearson's correlation matrix from the pairwise structure comparison matrix. This correlation matrix scaled, thresholded, and then converted into a weighted network from which SSI values can be calculated from each node. The median SSI value is plotted for many resamplings of nodes. **d)** To calculate bin variability, we convert each single-cell distance matrix into a thresholded matrix using a cutoff and then construct a network from each of those matrices with bins as nodes and contacts between bins as edges. We then calculate the bin similarity index for each bin in each single-cell and then take the standard deviation of the bin similarity index across the population as the bin variability.

associated with the global rotational and translational variations which might affect underlying statistical analysis. To resolve this issue, we convert each 3D structure into a 2D symmetric distance matrix by calculating the pairwise Euclidean distance between all the genomic bins of that chromosome structure. For example, if a particular chromosome has $m$ genomic bins and a total of $N$ copies originating from $N$ single cells, then we will have $N$ symmetric pairwise distance matrices each of size $m \times m$, where $d_{ij}$ represents the Euclidean distance between $i^{th}$ and $j^{th}$ genomic bins. Further, the spatial distance can be converted into contacts by applying a genomic resolution specific thresholding cutoff. However, for our analyses, we use distance matrices instead of contact matrices, unless otherwise explicitly mentioned.

## Generating chromosome structural landscapes using a statistical analysis technique

We next center all the single-cell matrices around zero by subtracting the median distance matrix from each of them, $h_{ij} = d_{ij} - median(d_{ij})$. Once we obtain the median centered single-cell distance matrices, we calculate the pairwise dot product between each of those single-cell matrices, such as, $s_{\alpha\beta} = \sum_{i,j} h_{ij}^{\alpha} . h_{ij}^{\beta}$, where $h^{\alpha}$ and $h^{\beta}$ are median centered distance matrices obtained from single cells ($\alpha, \beta \in \{1,...,N\}$), respectively. For each pair of single-cells, this step compares the deviation from the median between the two structures and yields a single number. In the case of two structures having similar deviations, this dot product would yield a strong positive value and vice versa. The pairwise dot product comparison information between all possible pairs of the single cells is then represented in a symmetric matrix format, termed as pairwise structure comparison matrix, where each element of the matrix represents the pairwise comparison information between a specific pair of chromosome structures. Next, we perform PCA on the pairwise structure comparison matrix and project the single-cell conformations in 2D space using the first two principal components (PCs). This PC projection is termed the chromosome structural landscape.

## Quantification of chromosome structural ensemble variation using a network-based similarity index

To quantify the variation associated with the chromosome structural ensemble, we perform a network-based similarity calculation on the pairwise comparison matrix (Fig 1C). Similar approaches have been used to study cellular network heterogeneity associated with cancer and cellular differentiation [46–48]. In this approach, we first calculate Pearson's correlation coefficient (PCC) between each pair of rows of the comparison matrix, where $c_{ij}$ denotes PCC between $i^{th}$ and $j^{th}$ rows of the pairwise similarity matrix. As PCC values range between -1 and 1, next we scale the PCC values in the range of 0 and 1, $w_{ij} = 0.5^{*}(1 + c_{ij})$. Once we have the rescaled pairwise PCC value matrix, we construct a weighted network from that where each node represents a single chromosome structure or conformation and the edges between them denotes rescaled PCC values. However, we only consider the edges having weights above a certain threshold, representing only connections between highly similar structures. This threshold is calculated by taking $50^{th}$ percentile of all the edge weights of the network (see Results for an analysis of the impact of this threshold choice). With the constructed network, we calculate the structure similarity index (SSI) of each node $S_i$, using a formula based on the Shannon-Jayne network entropy [46].

$$S_i = -\frac{1}{\log k_i} \sum_{j \in N(i)} p_{ij} \log p_{ij}$$

where $N(i)$ represents the neighborhood set of the node $i$ and $k_i$ denotes the cardinality of that set. $p_{ij}$ represents the proportionality of the edge weight between nodes $i$ and $j$ compared to all the edges associated with node $i$.

$$p_{ij} = \frac{w_{ij}}{\sum_{k \in N(i)} w_{ik}}$$

Network entropy is a commonly applied concept in the analysis of heterogeneity vs. continuity in biological networks [49]. Conceptually, entropy measures the degree of randomness in a system, or the extent to which all possible outcomes are sampled. Entropy in DNA sequence analysis measures whether all 4 DNA bases are represented uniformly in a motif position (high entropy) or some bases are strongly preferred in certain positions (low entropy) [50,51]. Network entropy in signaling or gene expression networks captures whether the network is more continuous or modular [46,52,53]. Similarly, here this metric will reveal whether the landscape of structures is highly connected with all intermediates sampled vs. more heterogeneous with disconnected structures. This index $S_i$ is bound between 0 and 1. In our system, the resulting values are compressed near a value of 1, so for easier visual presentation of values, we transform them by subtracting 0.99 from each SSI and then multiplying by 1000. For our presented analyses, we calculate the median SSI across all nodes in the landscape and call this the ensemble SSI. Specifically, to normalize sample size between conditions, we iteratively resample a subset of nodes from each condition, then plot the distribution of median SSI values from the resampled node sets. Based on the distribution of the median SSI values, we characterize the heterogeneity of the chromosome structural landscape. An SSI distribution with higher mean is considered as having a smooth continuous structural landscape while a distribution with relatively lower mean would have more discrete, disconnected structures. Interpretation of the SSI distribution trends have been discussed in detail in the **Results** section.

### Calculating chromosome structure variability for each genomic bin

To characterize the variability of structure at each genomic bin along the chromosome, we use a modified version of the network-based structural similarity calculation approach, discussed in the previous subsection (Fig 1D). In this step, for each single-cell structure, we first construct a network based on its distance matrix, where each node is a single genomic bin and the edges between them represent the spatial distances. We then threshold the network using an experimentally-derived distance cutoff (330 nm for 30 kb resolution [41] and 500 nm for 50 kb resolution [38]), where edges with spatial distance higher than the cutoff are discarded from the analysis. Next, we calculate the bin similarity index of each of the nodes (bins) in the network using the same formula discussed in the previous section, except here using genomic bins as nodes rather than whole single-cell structures and spatial distances as edge weights rather than structure correlation coefficients. We now have a bin similarity index value for each genomic bin in each single cell structure. To quantify the variability of structural positions occupied by a given bin across all structures, we then calculate the standard deviation of bin similarity index values across all single cell structures for a given bin. We call this measure the bin variability. A higher standard deviation corresponds to a more dynamic or variable genomic bin.

### Characterization of chromatin states using epigenetic profiles

In order to characterize chromatin states of the IMR90, K562 and A549 cells, we downloaded a total of eleven histone marks–H3K4Me1, H3K4Me2, H3K4Me3, H3K9Ac, H3K27Ac, H4K20Me1, H3K36Me3, H3K27Me3, H3K9Me3, H2A.Z and DNase for all the three cell types

from ENCODE repository (S1 Table) [54]. Next, we used chromHMM software [55] to annotate genomic regions of desired resolutions into five chromatin states using a combinatorial pattern of the eleven histone marks. Based on the combinatorial histone marks, we defined chromatin states as–Active state 1, Active state 2, Inactive Poised, Repressed Polycomb and Heterochromatin (S1 Fig). Briefly, if an inferred state shows strong enrichments for active marks such as H3K4Me1/2/3, H3K27Ac, H3K9Ac and DNase, we term that state as an Active state. Further, depending on the level of enrichment of those different active marks, we characterize them into Active state 1 and Active state 2. We assign the Heterochromatin label to the state for which we observe strong enrichment of H3K9Me3 and for the Repressive Polycomb assignment, we look for strong H3K27Me3 and a mix of H3K9Me3 and H4K20Me1 marks. And the rest of the states are assigned the Inactive Poised level.

## Polymer simulation of chromatin segments based on underlying chromatin states

To test the contribution of epigenetic states to chromosome structure heterogeneity, we perform polymer simulations of a specific chromatin segment with the interactions defined based on their epigenetic states. In recent years, polymer simulation-based studies have shown how the compartmental profile or the underlying epigenetic state of the genomic regions can give rise to the population-averaged structural organization of chromatin at different resolutions [9,56,57]. Here, we represent a ~ 1.95 Mb chromatin region as a chain of beads using a bead-spring polymer model. The model system consists of 1950 beads where beads represent genomic regions of 1 kb. Each bead is assigned a specific chromatin state as described above. In addition to that, we also model loops within the specific chromatin regions based on publicly available high-resolution Hi-C derived chromatin loop sets from Rao $et$ $al.$ [58]. For each loop, we connect the loop anchors using a harmonic constraint. The potential energy function of the polymer model is described below,

$$U_{\mathrm{ref}}(r) = U_B(r) + U_P(r) + U_L(r) + U_W(r)$$

where $U_B(r)$ corresponds to bond stretching potential, modeled by finite extensible nonlinear elastic potential given by,

$$U_B(r) = -0.5 \mathrm{K}_S R_0^2 \ln\left[1 - \left(\frac{r}{R_0}\right)^2\right]$$

where $\mathrm{K}_S = 30.0 \frac{\epsilon}{\sigma^2}$ is spring constant and $R_0 = 1.6\sigma$ is the equilibrium bond length.

We use Lennard-Jones potential to model steric repulsion and attraction between the beads of different chromatin state with different parameters. For example, the potential between a bead of chromatin state $A$ and that of $B$ is represented below,

$$U_P(r) = 4\epsilon_{\mathrm{AB}}\left[\left(\frac{\sigma}{r}\right)^{12} - \left(\frac{\sigma}{r}\right)^6\right]$$

where $\epsilon_{\mathrm{AB}}$ denotes interaction strength between bead types $A$ and $B$ (S2 Table) and σ refers to the size of each bead.

In case of loops, we use harmonic potential between each of the loop anchors $U_L(r)$, represented by,

$$U_L(r) = K_L(r - R_0^L)^2$$

where $\mathrm{K}_L = 300 \frac{\epsilon}{\sigma^2}$ is spring constant and $R_0^L = 1.2\sigma$ is the equilibrium loop bond length.

Finally, in order to model the repulsive interaction between the chromatin beads and the simulation box boundaries, we use Lennard-Jones potential

$$U_W(r) = 4\epsilon_W \left[ \left(\frac{\sigma}{r}\right)^{12} - \left(\frac{\sigma}{r}\right)^{6} \right]$$

where $\epsilon_W = 1.0$ denotes the interaction strength between boundary walls and chromatin beads irrespective of their types.

The LAMMPS molecular dynamics package is used to perform polymer dynamics simulations [59]. All the simulations are performed in terms of Lennard-Jones reduced units with $\tau$ for time, $\sigma$ for length and $\epsilon$ for energy. To mimic the viscoelastic nature of the nucleoplasm, Langevin dynamics with a damping coefficient of $\gamma = 1.0\tau$ is used to maintain the temperature at $T = 1.0$. We then run the simulation for a total of 1,000,000 steps with a timestep of $dt = 0.000001\tau$. Followed by that, the timestep dt is increased to $0.01\tau$ and a production run of a total 2,000,000 steps is performed and the conformations are saved at intervals of 1,000 steps. We repeated this process for 20 times, each time with different initial conformation and random seeds.

## Initial conditions and Equilibration

The chromatin fiber is initialized as a self-avoiding worm like chain using the python package PolymerCpp [60] and is placed in a box of size $L = 35\sigma$ with fixed repulsive boundaries. We then allow the polymer to equilibrate inside the box using four consecutive phases. In the first phase, a soft potential is used to remove overlap in the polymer for 400,000 steps with a $dt = 0.01\tau$.

$$U_S(r) = A \left[ 1 + cos\left(\frac{\pi r}{R_0^S}\right) \right]$$

where $A$ is pre-factor, varied gradually from 0 to $100\epsilon$ and $R_0^S = 2^{1/6}\sigma$.

For the rest three subsequent phases, we switch to potential to Lennard-Jones potential and run the simulation for 400,000, 500,000 and 500,000 steps with timesteps $0.000001\tau$, $0.0001\tau$, and $0.01\tau$ respectively.

$$U_G(r) = 4\epsilon_G \left[ \left(\frac{\sigma}{r}\right)^{12} - \left(\frac{\sigma}{r}\right)^{6} \right]$$

where $\epsilon_G = 1.0$ denotes the interaction strength between chromatin beads irrespective of their types.

## Loop extrusion simulation of chromatin segments based on underlying RAD21 binding sites

To test the contribution of loop extrusion dynamics to the variability present in chromosome structural ensemble, here we perform loop extrusion polymer simulations of a specific chromatin segment according to the method described in Fudenberg *et al.* [2]. For these simulations, we define the domain boundaries based on RAD21 peaks present in the desired chromatin segment. Here, we model a 2 Mb spanning chromatin region as a 2000 beads monomer chain where beads represent genomic regions of 1 kb. All the simulation parameters are kept same between different cell types except the position of the boundary elements (here RAD21 peaks) and their strengths. The strength of boundary elements is defined in terms of boundary permeability parameter in the models which represents the boundary insulation.

For each cell-type, these parameter values are obtained by calculating the relative strength of the RAD21 peaks. The RAD21 peaks for the IMR90 and K562 were downloaded from the ENCODE repository [54].

### Analysis of nucleolus-associated domain (NAD) mapping data

In order to characterize the nucleolar association of the IMR90 chromosomal regions, we downloaded the NAD mapping data of the proliferating IMR90 cells from Dillinger *et al*. [61]. This microarray data contains log2 normalized ratio of nucleolus-associated and genomic DNA at specific probe locations. For our analysis, we bin that log2 normalized ratio data at 10 kb resolution using multiBigwigSummary functionality of the deepTools software [62].

## Results

### Chromosome structural landscape reveals major features of the structural ensemble

We first apply our statistical analysis technique to the single-cell chromosome structures measured for a 2 Mb region of IMR90 chr21 (28–30 Mb) (Fig 2A) and produce a chromosome structural landscape (Fig 2B). We observe that the first major source of variation corresponds well to the compactness of the chromosome structures, which can be quantified using the radius of gyration, $R_g$ (S2A Fig). We calculate the $R_g$ of the single-cell structures from their experimentally obtained 3D conformations. When we compare the $R_g$ values of the structures with their PC1 values (from here on we refer to the PC1 projection as "PC1"), we observe that the structures with more negative PC1 projections are highly compact with smaller $R_g$ values while structures with more positive PC1 values are less compact with higher $R_g$ scores. With increasing $R_g$, the structural divergence also increases, with individual structure points becoming spread further along PC2 and also more spread out from each other.

Along PC2, we observe that the structure ordering follows a progressive reorganization of the domains within the region and captures several subclusters of structures (Fig 2C). As we move from negative to positive PC2, we observe a shifting of a boundary (weakening of one boundary and gaining of another) that leads to the change in the structure of the domain (S2B Fig). This phenomenon can be seen in different regimes of $R_g$, showing the existence of domain reorganization within both compact and open structures (S2B Fig).

When the same approach is applied to different chromosomes and cell-types at different resolutions, we found a consistent ordering of the dynamic modes of the structural landscape (S2C, S2D, S3 and S4 Figs). Although the major dynamic variation remains the $R_g$ ordering for all the different situations, the reorganization of domains along PC2 is dependent on the length of the genomic region as well as the local structural properties.

### Ensemble structure similarity index captures cell-type- and chromosome-specific features of the chromosome structural ensemble

To compare the single cell structural heterogeneity between different cell types, we constructed a joint structural landscape using the 28–30 Mb chr21 region data from three cell types– IMR90 (lung fibroblast), K562 (leukemia) and A549 (lung adenocarcinoma) collectively. In this joint structural landscape we can visualize the cell-type-specific variation present in structural ensemble (Fig 3A). To quantify the differences between cell types, we divide the landscape in 10 equally sized regimes along PC1 (since PC1 correlates with $R_g$, different PC1 regimes correspond to different $R_g$ values) and calculated the proportion of different cell types within each regime. We observe that K562 shows a higher proportion of chromosome

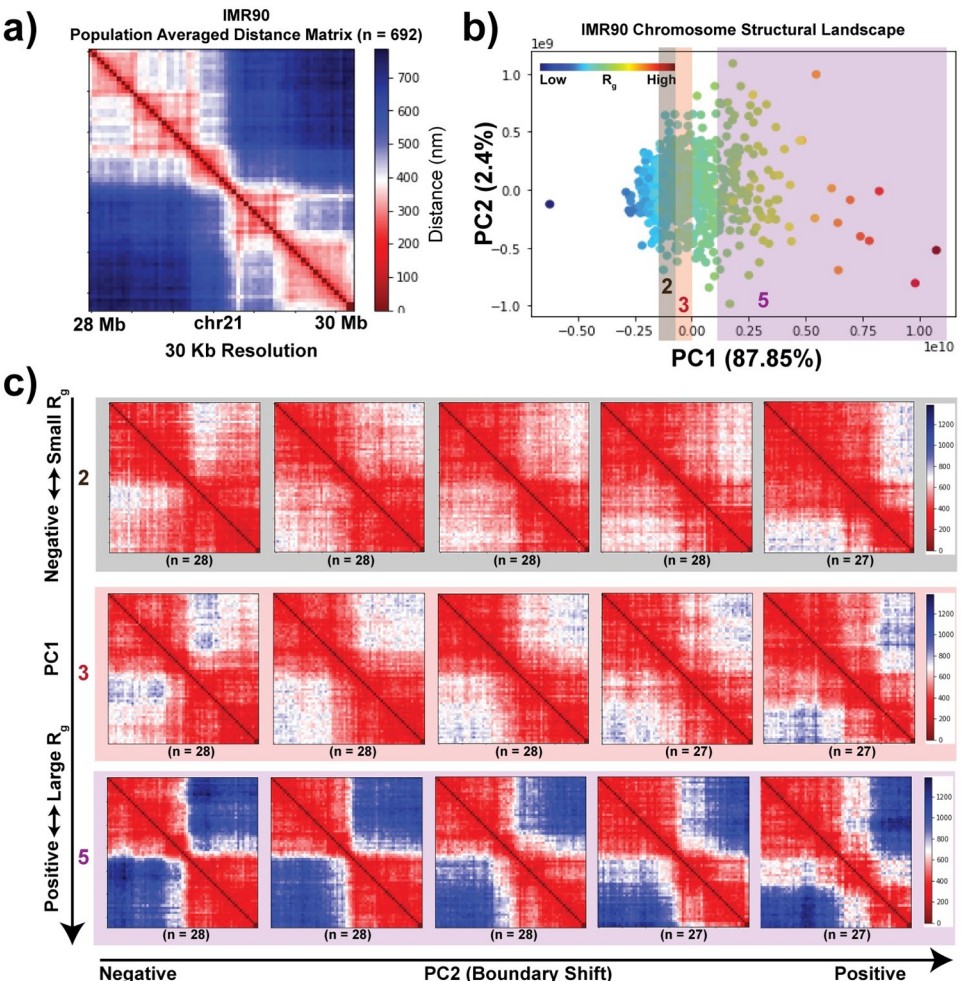

**Fig 2. The chromosome structural landscape orders single-cell structures on a continuum based on their radius of gyration followed by domain reorganization. a)** Population-averaged distance map of IMR90 chr21:28–30 Mb at 30 kb resolution. **b)** Chromosome structural landscape of IMR90 chr21:28–30 Mb chromatin region, generated as described in the **Methods** section. Each point represents a single-cell structure. Points are colored according to the radius of gyration ($R_g$) of the corresponding structures (dark blue–smaller $R_g$ and dark red–larger $R_g$). The percent of variation explained by each PC is shown in parentheses. **c)** Structures are first divided into five groups based on the PC1 ordering (which correlates highly with $R_g$). Structures belonging to groups 2, 3 and 5 are shown. Within each group structures are further divided into five subgroups based on their PC2 ordering. Average structures within each subgroup are shown.

structures in lower $R_g$ regimes and has an overall more compact structural ensemble compared to A549 and IMR90 (Fig 3A and 3B).

While the relative $R_g$ shift between cell types is straightforward to observe and quantify, other features of the chromosome structural landscape are more difficult to compare visually. Therefore, we devised a network-based measurement to quantitatively compare the structural landscapes, as discussed in the **Methods** section. According to this approach, a network with similar edge weights will exhibit a higher ensemble SSI value and vice-versa. In terms of the chromosome structural ensemble, a set of structures that smoothly transition across the entire conformational space will have a highly connected structural landscape and thus high ensemble SSI (Fig 1A, blue line). On the other hand, if structures coalesce around discrete states (Fig 1A, red line), the discontinuity in the structural ensemble will be higher, and the resulting

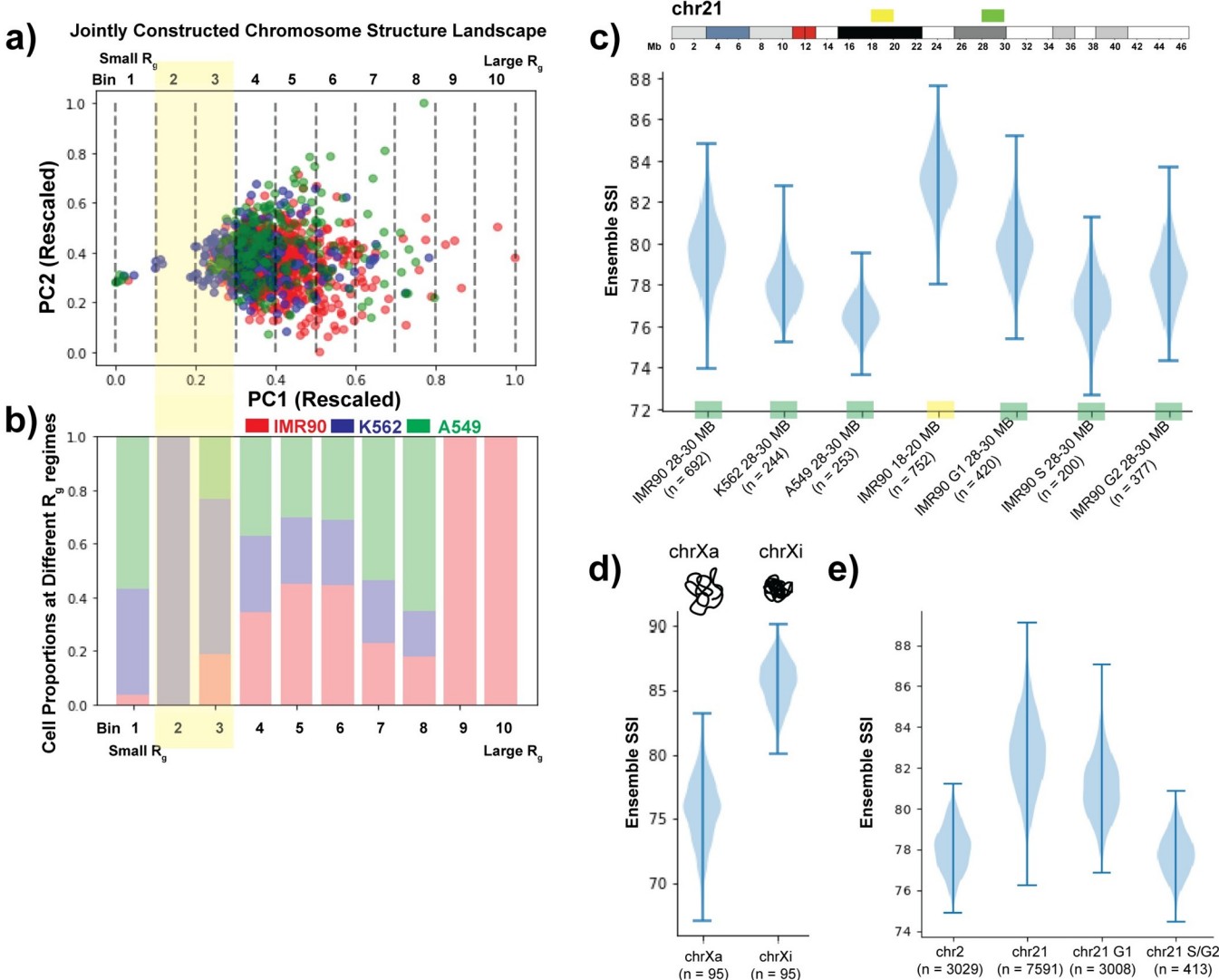

**Fig 3. Network based similarity score quantifies cell-type- and chromosome-specific variability present in chromosome structural ensemble at different resolutions. a)** Jointly constructed chromosome structural landscape of genomic region chr21:28–30 Mb for cell types IMR90, K562 and A549. The only difference between individual and joint landscape is that all the different cell-type structures are considered together while calculating pairwise structure comparison matrix. Points are colored based on the cell types. Further, the landscape is divided into ten equal sized bins (or regions) based on PC1 values, where the bins on the left represent lower $R_g$ regions and bins on the right are with higher $R_g$. **b)** Relative proportion of IMR90, K562 and A549 single-cell structures present in each bin represented in a). **c)** Chromosome structural ensemble SSI distributions for different 2 Mb chromosome regions from different cell-types and conditions, obtained from Bintu *et al.* [37]. The SSI value of the structural ensemble is calculated using a network entropy based approach. For each structural ensemble (size of ensemble indicated as n), 200 single-cell structures are selected randomly (except IMR90 S 28–30 Mb where we select 150 structures for resampling), and the median of the SSI values of the nodes are calculated. This step is repeated 1000 times and values are plotted as a distribution. **d)** Chromosome structural ensemble SSI distributions for active and inactive IMR90 chrX at TAD resolution, obtained from Wang *et al.* [36]. 50 single-cell structures are selected randomly each time for resampling. **e)** Chromosome structural ensemble SSI distributions for different IMR90 chromosomes and conditions at higher resolution, obtained from Su *et al.* [38]. Here, for resampling 300 single-cell structures are selected randomly each time.

network will have a larger variation in of edge weights, and a low ensemble SSI score will be produced.

We first applied the network-based similarity quantification approach to structures of 2 Mb regions of chr21 (30 Kb resolution) in different cell-types and conditions obtained from Bintu *et al.* [37]. There is variation in the sample size (i.e., the number of conformations) obtained for different experiments, so we relied on a resampling approach to make the comparison

consistent. Based on this approach, when we compared the 28–30 Mb genomic region from three different cell-types analyzed above, we find that IMR90 has the highest ensemble SSI, followed by K562 and then A549 with the lowest ensemble SSI (Fig 3C). Thus, this region in IMR90 cells has the highest continuity across different structures while the corresponding distribution in A549 cells exists as a relatively discrete set of structures.

To evaluate our interpretation of the ensemble SSI result in an orthogonal fashion, we performed hierarchical clustering of the 3D structures obtained from IMR90 and K562 cells based on a structure similarity metric, Q, as represented in Cheng *et al.* (S5 Fig). At the same threshold, both cell types showed the same number of structure clusters. However, in IMR90, the vast majority of structures fall into one of these clusters, corresponding to our high ensemble SSI value (S5A and S5B Fig). In K562, the structures are more evenly split into the different clusters, corresponding to our low ensemble SSI value (S5C and S5D Fig). Thus, our ensemble SSI metric indeed captures the degree of continuity within the structural ensemble. By using ensemble SSI rather than a hierarchical clustering approach, we avoid the need to arbitrarily select a cutoff or number of clusters. We further assessed the dependence of the ensemble SSI score on the number of cells measured (S6 Fig). We observed that with an increase in the number of cells, the difference in ensemble SSI between cell-types follows the same trend, but the absolute difference becomes smaller (S6B Fig). It is important to note that as we increase the number of cells, we are including more low-quality measurements (cells with more missing data) and this may also contribute to a broader ensemble SSI distribution. We then examined how robust the ensemble SSI distribution is to changes in the correlation matrix cutoff chosen (S6C Fig). Using a cutoff lower than our initially chosen 50th percentile produces similar results, but increases variability as more non-significant connections between structures are included. A high cutoff, on the other hand, can obscure the results by breaking a heterogeneous structure landscape into too many disconnected networks. Within each sub-network, there is high continuity, so this results in an artificially high SSI being reported for landscapes that are in fact heterogeneous. We conclude that a 50th percentile correlation matrix cutoff excludes noisy connections without creating disconnected subnetworks (see S6C Fig for extended discussion). As a final technical test of our metric, we repeated the ensemble SSI analysis after removing extremely compact or large structures from each cell type to see whether those extreme conformations are disproportionately affecting the results (S7 Fig). We found that those extreme structures actually have little influence on the ensemble SSI calculation. Overall, we conclude that our ensemble SSI calculation is robust.

We next compared two different genomic regions of IMR90 chr21 using the ensemble SSI distribution. We found that although both are from same cell-type and same chromosome, the 28–30 Mb region has lower ensemble SSI values compared to 18–20 Mb region (Fig 3C). This difference indicates that the latter region has a higher degree of continuity in its structural ensemble. This 18–20 Mb region lies entirely in the B compartment (putative inactive, gene poor regions) while the 28–30 Mb region contains a majority of A compartment regions mixed with B compartment regions. To further investigate the potential connection between compact, inactive states and the continuity of the structural landscape, we compared the ensemble structural similarity of the active vs. inactive chrX structural ensembles (440 Kb to 7.8 Mb) (Fig 3D). We found that chrXi has higher ensemble SSI values compared to its active counterpart. This suggests that the chromosome with a highly compact global structure has a more continuous structural landscape compared to the more specific and discrete states of the open and active chromosome. (S8A, S8B and S8C Fig) [63].

When we analyzed the structural variation of the entire chr2 and chr21 q-arm at high resolution [38], we observed that chr21 has a higher ensemble SSI compared to chr2 (Fig 3E). To test whether the difference in length between chr2 and chr21 contributed to differences in the

ensemble SSI values, we calculated the ensemble SSI of regions of different lengths selected from chr2. We observed that as the length increases the ensemble similarity decreases (S8D Fig). This effect likely stems from the increased number of possible conformations that a longer polymer can adopt, which corresponds to a sparser coverage of conformational space. We also found that ensemble similarity is lower in S/G2 phase compared to G1 phase for chr21. This result is consistent with the ensemble SSI values of 28–30 Mb IMR90 chr21 region structural ensembles from different cell-cycle stages, where the G1 phase has highest ensemble SSI followed by S and G2 phases, as shown in Fig 3A. One major difference between the chromosomes in the phases of the cell cycle is the addition of tethers between sister chromatids in S and G2 compared to G1 [64]. Increased constraints on the chromosome structure could then lead to more distinct structures and lower structural ensemble similarity.

## Contribution of epigenetic properties to the cell-type-specific variation of chromosome structural ensemble

Building on the observation that regions with different A/B spatial compartment identity had different structural landscape SSI, we further examined the influence of local epigenetic state on the resulting structural landscape. Recent studies have shown the association of epigenetic states with the ensemble genome organization [28, 29, 65]. Fig 4A shows chromatin states along the 28–30 Mb genomic region in the three different cell types–IMR90, K562 and A549 (detailed steps in the **Methods** section). We observe that this region in IMR90 mostly has active marks while in K562 the region is mixed between repressive polycomb and heterochromatin marks as well as active marks. The A549 region shows a mixture of active, repressive, and poised marks. By qualitatively comparing this chromatin state data with the ensemble SSI distributions for these three cell types, we observe that the cell type with the more uniform epigenetic state (IMR90) has the highest ensemble SSI while cell types with mixed epigenetic states exhibit lower SSI.

In addition to the overall dynamic properties of the region, we sought to quantitatively compare local genomic bin epigenetic state with the degree of structure fluctuation experienced by that bin. We first calculated the bin variability for each cell type (**Methods**) which represents the structure variability of a particular genomic region across the population of cells (Fig 4B). We do not see a direct bin-by-bin correlation between epigenetic state and bin variability. Instead, we observe that a higher proportion of active states within the region tends to correspond to higher bin variability across the region while a larger proportion of inactive states, even if focused in one part of the locus, corresponds to restricted bin variability across the whole region. In this comparison, it is important to remember that we are comparing a population averaged epigenetic state with single cell structural fluctuation. It is possible that additional relationships could become evident if single cell variation in epigenetic state were considered.

To test our hypothesis that epigenetic state differences contribute to cell-type-specific variation in the chromosome structural ensemble, we performed a set of polymer simulations of the desired genomic region of 28–29.5 Mb at a resolution of 1 kb, as described in the **Methods** section. We generated the models for IMR90 and K562 separately by assigning interactions between the polymer beads based on their specific chromatin states (S9A Fig). Once we obtained the 3D conformations from the simulations, we constructed pairwise Euclidean distance matrices for each conformation and further quantified the structural ensemble using our structural similarity approach. For the resampling purpose, we combined all the snapshots from 20 independent simulations and then selected 500 conformations randomly from this set 1000 times. When we compared the simulated IMR90 ensemble SSI distribution with

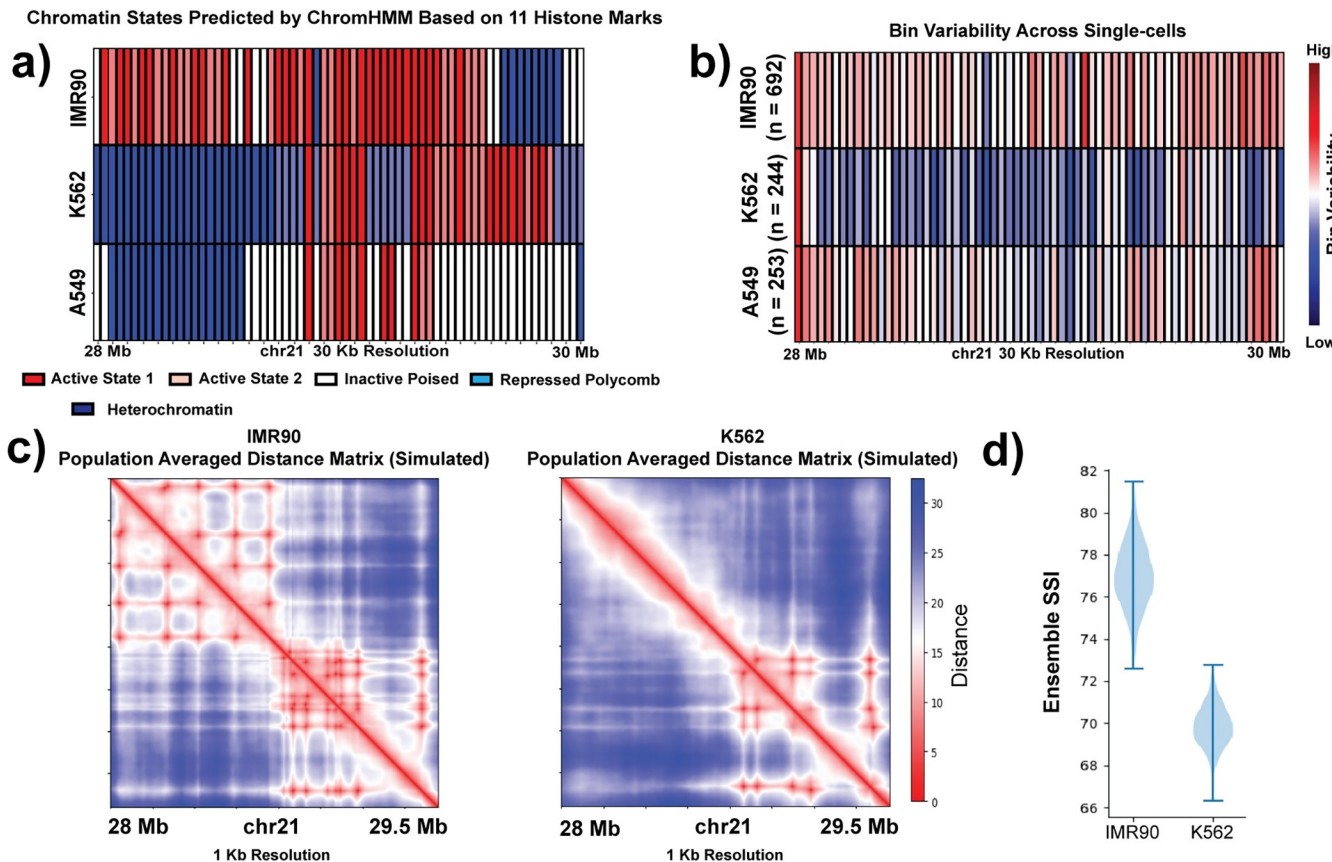

**Fig 4. Epigenetic properties (histone modifications) contribute to the cell-type-specific variation in the chromosome structural ensemble. a)** Chromatin states of chr21:28–30 Mb genomic region from three different cell-types–IMR90, K562 and A549 at 30 kb resolution. The chromatin states are inferred based on 11 different histone marks for each cell-type by ChromHMM tool. **b)** Bin variability of the chr21:28–30 Mb region from IMR90, K562 and A549, where a higher value represents higher degree of structural variation of that genomic position across the population and vice versa. **c)** Population-averaged distance maps of IMR90 and K562 chr21:28–29.5 Mb genomic region from polymer simulations (see Methods) at 1 kb resolution. **d)** Chromosome structural ensemble SSI distributions for chr21:28–29.5 Mb from simulated IMR90 and K562. For resampling, 500 single-cell structures are selected randomly each time.

simulated K562 data, we found that the K562 structural ensemble shows a lower similarity, which is the same trend observed in our analysis of the Bintu *et al*. imaging data (Figs 3C and 4D). Since the only difference between cell types in these simulations was the encoded epigenetic state of the polymer beads, this result corroborates the idea that the epigenetic state is one factor that can contribute to the variation in chromosome structural ensemble.

## Structure heterogeneity properties are associated with epigenetic state and boundary positions

We further evaluated the contribution of epigenetic factors to structural heterogeneity by analyzing single-cell structures of six genomic regions, each 2 Mb in length, obtained from IMR90 chr21 whole q-arm imaging data (Fig 5A) [38]. For simplicity, the regions will be denoted as S1 to S6 based on their starting genomic coordinates. The regions S1 (chr21:18–20 Mb) and S2 (chr21:28–30 Mb) were selected in order to match the two IMR90 genomic regions data analyzed in Bintu *et al*. The three regions S3, S5 and S6 were selected based on their epigenetic state similarity with epigenetic state of the chr21:28–30 Mb region in K562 cells (S9B Fig). To evaluate epigenetic state similarity, we designed a sliding window-based approach where we move a window of size 40 (2 Mb genomic region with 50 kb resolution) over the whole q-arm

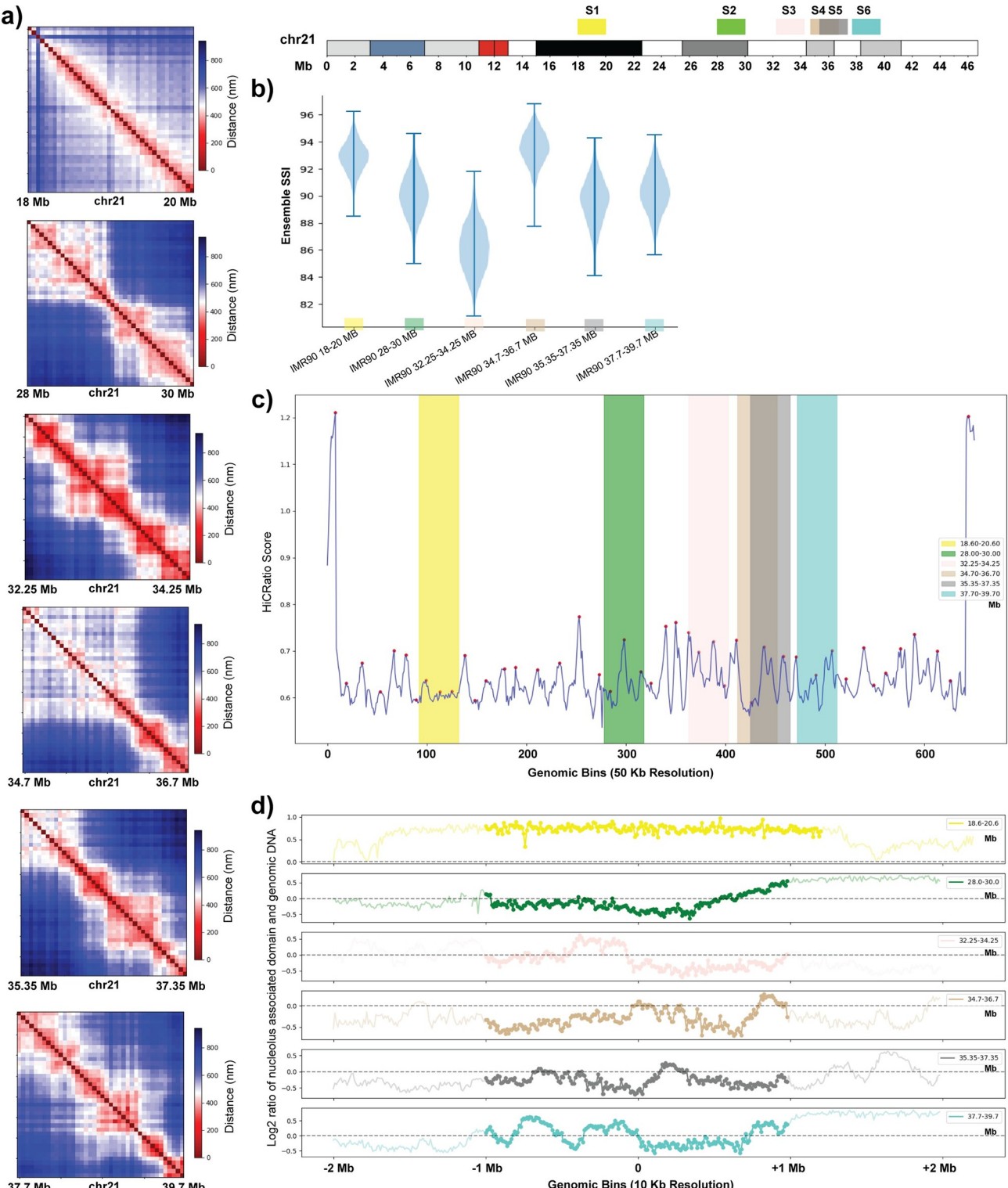

**Fig 5. Evaluation of structural variation of regions with different epigenetic, genomic, and structural properties. a)** Population-averaged distance maps of 6 different genomic regions of IMR90 chr21 at 50 kb resolution obtained from Su *et al.* [38]. Locations of these 6 regions are indicated on the top right chromosome ideogram and labeled with colors used throughout the figure. **b)** Chromosome structural ensemble SSI distributions for the same 6 regions of IMR90 chr21. For resampling, 300 single-cell structures are selected randomly each time. **c)** HiCRatio score of each genomic bin of chr21 q-arm at 50 kb resolution (calculated with a 300 kb window size). The red points represent the boundary positions inferred by HiCRatio. **d)** Nucleolus

association level of the same 6 regions of IMR90 chr21 [61]. For each genomic region, the 1 Mb upstream and downstream regions are also shown. A value greater than 0 represents higher levels of nucleolus association and vice versa. Colors in each panel correspond to chromosome locations indicated above panel b.

of chr21 and for each window compare the IMR90 epigenetic state with the K562 28–30 Mb region epigenetic state using binary comparison (S9C Fig). However, as the resolution of the Bintu *et al.* data (65 bins per 2 Mb region) is different from the Su *et al.* data (40 bins per 2 Mb region), we take the least common multiple of the two bin numbers and expanded each data to 250 bins by repeating each bin element the required number of times. Here, a higher comparison value shows higher similarity with the K562 epigenetic state. Among the three regions, S3 has the highest score, S6 has the second highest and S5 has the third highest one (S9C Fig). The region S4 was selected randomly.

When we quantified the variation in the structural ensemble of these six regions, we first observed that the S1 and S2 regions have ensemble SSI distributions that are consistent with the results from the same regions in the Bintu *et al.* data (Fig 5B). Next, we found that the S3 region has the lowest ensemble SSI distribution among all the six regions which is also consistent with the low ensemble SSI of the Bintu *et al.* K562 region. In addition, the ordering of S6 and S5 are also consistent with their similarity with K562 epigenetic state. Finally, the S4 region, which has a very low similarity score with K562 state, exhibits a very high ensemble SSI distribution. Again, this analysis supports the idea that epigenetic patterns contribute to the degree of variation present in the chromosome strucural ensemble.

In addition to the epigenetic states, we also identified domain boundaries along the whole q-arm of chr21 using the HiCRatio approach (with a 300 kb window size) [66]. Since this boundary calling software is designed for contact data, we first converted the single-cell chromosome structural distance data into contact data using a cutoff of 500 nm, as previously reported [38]. We found that regions with a higher number of strong domain boundaries exhibit lower ensemble similarity and vice versa (Fig 5C). For example, the S3 region, which has the lowest ensemble SSI, has three strong domain boundaries and one weak. On the other hand, regions like S1 and S4 which exhibit higher ensemble SSIs have fewer or weaker domain boundaries. This effect is visible across the entire chromosome, in that the average boundary strength in a given region is negatively correlated with the ensemble SSI of that region (S9 and S9E Fig).

Finally, we asked whether chromosome association with the nucleolus is associated with changes in the chromosome structural ensemble SSI (Fig 5D). The NAD data source and processing steps are described in the **Methods** section. We observe that S1 and S4, the regions with the highest structural ensemble similarity, are the most uniform in their NAD state: S1 is completely associated with the nucleolus, while S4 has almost no association with the nucleolus. As seen with epigenetic state, here we find that regions with uniform nucleolus association exhibit a higher continuity level in the structural ensemble. In contrast, S3 has the lowest ensemble SSI and also alternating levels of nucleolus association in the first half of the genomic region, followed by an unconstrained region. Thus, variation in constraints across the region is again associated with specific distinct structures and thus a lower ensemble similarity.

## Loop extrusion alone does not reproduce experimentally detected chromosome structure variation patterns

Above, we observed that chromosome regions with a higher number or strength of domain boundaries showed less continuous structural landscapes and lower ensemble SSI values. Boundaries detected by approaches like HiCRatio can result from cohesin-mediated loop extrusion that is periodically blocked by factors such as CTCF [67], as well as from the borders

between A and B compartment regions of chromatin. We next therefore asked whether the loop extrusion process, blocked by different numbers of boundaries, would be expected to generate the patterns of structure variation we observe. We simulated loop extrusion using the chr21:28–30 Mb segment using experimental cohesin binding data from K562 and IMR90 (see Methods). This region has fewer boundaries in K562 (likely due to the portion of the region that is heterochromatic in this cell type). The structure landscapes resulting from a single simulation run show that the cell type with more loop extrusion boundaries (IMR90) would be predicted to have a lower ensemble SSI than the cell type with fewer boundaries if only loop extrusion were at work (S10A and S10B Fig). Once we start the simulation from several different initial conditions, this difference is diminished (S10B Fig) due to variation in structure between different runs. While this result matches our expectations that more boundaries associate with lower ensemble SSI, it does not agree with the experimental observations from these cell types, in which IMR90 actually has the higher ensemble SSI. This suggests that epigenetic state-specific interactions must be considered along with loop extrusion to explain the chromosome variability observed.

### The effect of cohesin depletion on structural heterogeneity depends on the underlying epigenetic state of the chromosome region

To further examine the effect of cohesin on the chromosome structural landscape, we analyzed chr21 single-cell structural data from HCT116 cells in wildtype (WT) and cohesin depleted (+-Auxin) conditions [37]. This analysis included two different genomic regions (chr21:28–30 Mb and chr21:34–37 Mb) with different underlying structural properties. The jointly constructed chromosome structural landscape for both genomic regions shows that along PC1 the structures are again ordered based on their $R_g$ (S10C and S10D Fig). Thus, the chromosome structural landscape approach orders single-cell structures based on their $R_g$ irrespective of cell-type, genomic regions, and perturbation condition. The WT and cohesin-depleted structural landscapes from the first genomic region mostly overlap with a slight bias for the +Auxin structures in the higher $R_g$ regime (Fig 6C). This suggests that removal of cohesin makes the structures less compact to some extent, as previously observed [68].

On the other hand, for the second region, the jointly constructed landscape shows a shift upwards along PC2 along with the shift towards the large $R_g$ regime for cohesin-depleted structures (Fig 6D). In addition, we found that after cohesin removal, both genomic regions exhibit more structural variation at the genomic bin level according to BSI standard deviation (S10E and S10F Fig). When we examine the structural ensemble SSI of these two regions, we see apparently conflicting effects of cohesin removal: the ensemble SSI is decreased in the 28–30 Mb region but mostly unchanged in the 34–37 Mb region (Fig 6E). Note that the initial ensemble SSI of these regions in the WT follows the expected pattern for regions of different lengths. We observed that this divergent ensemble SSI effect corresponds to a difference in the underlying compartment state of the regions. The 28–30 Mb region is divided into A and B compartment segments (which in earlier examples tends to correspond to lower SSI) (Fig 6A) while the 34–37 Mb region is all in the A compartment with no compartment boundaries (which in earlier examples tends to correspond to higher SSI) (Fig 6B).

## Discussion

As the number and quality of single cell chromosome structural datasets continues to increase, a diverse toolbox of methods to capture properties of chromosome structure variation is needed. Our statistical analysis of single cell chromosome structures allows us to characterize both the major modes of chromosome structure fluctuation and the properties of the structure

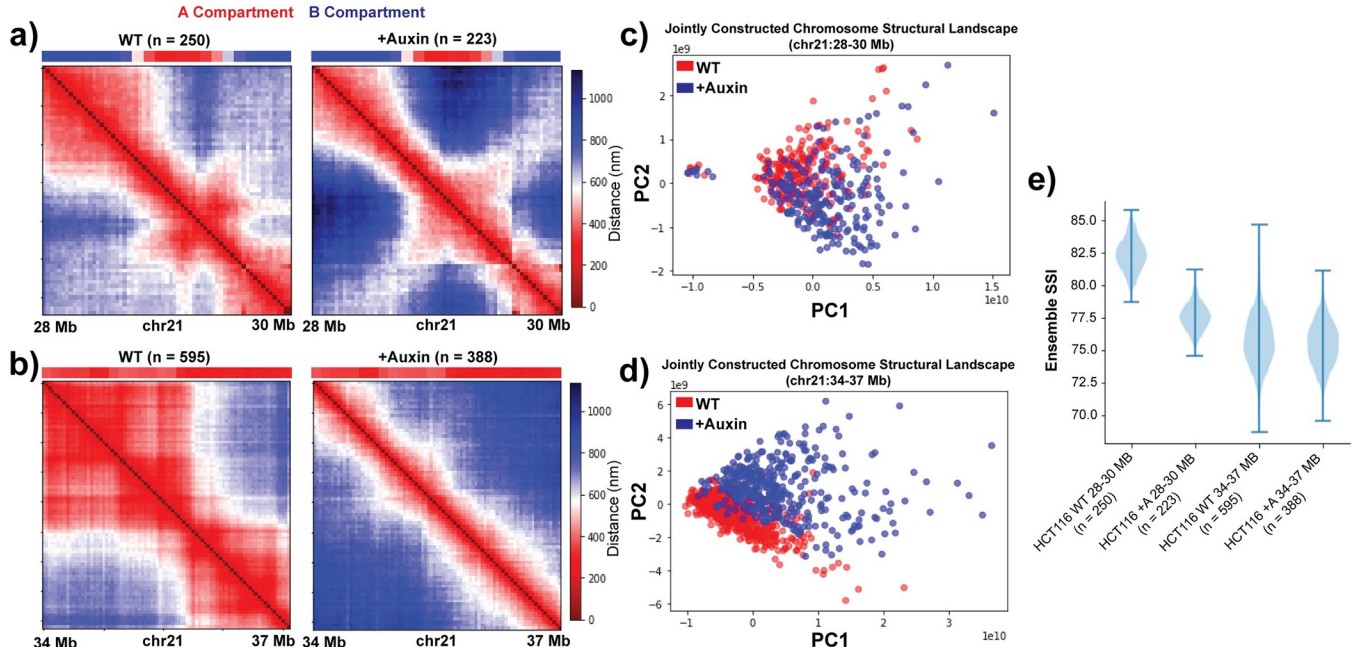

**Fig 6. Cohesin depletion alters chromosome structural heterogeneity depending on underlying epigenetic state constraints. a,b)** Population-averaged distance maps of HCT116 chr21:28–30 Mb (a) or chr21:34–37 Mb (b) at 30 kb resolution for wild-type (WT) and cohesin-depleted (+Auxin) conditions based on data obtained from Bintu *et al.* (37). Compartment status from Hi-C data at these locations shown above each matrix. **c,d)** Jointly constructed chromosome structural landscapes of chr21:28–30 Mb (c) or chr21:34–37 Mb (d) of HCT116 for WT and +Auxin conditions. **e)** Chromosome structural ensemble SSI distributions for different local chromosome regions of HCT116 for WT and +Auxin conditions. 200 single-cell structures are selected randomly each time for resampling.

fluctuation landscape. While some approaches focus on classifying single cell structures into discrete clusters, other recent work has shown that chromosome structures can exist along a continuum [25,30,39,41,42]. Our analysis does not require a decision to be made about how many different clusters of structures exist, but instead can capture the overall types of structure variations and can measure the degree of continuity between the structures.

For all observed length scales of chromosome structures, from local TAD scale to whole chromosome arms, we find that the primary type of single cell structure variation is the overall compaction of the region. For each structure assayed, there exist some cells in which the region being examined is very compact and others in which it is much more extended. This overall size fluctuation is true of both active and inactive regions, though inactive regions and the inactive X chromosome show a shift toward smaller sizes overall. However, we find that this overall region compaction is largely independent of a secondary fluctuation mode: the reorganization of domains. No matter how compact or extended the overall region is, we observe a similar progressive reorganization of the domain boundaries as a secondary type of structure variation.

By quantifying the network-based similarity properties of the structural ensemble, we can measure how discrete vs. continuous the structures adopted by a certain chromosome region are. Combining all of our observations of ensemble SSI in different regions and conditions, we propose that the ensemble SSI of a structural ensemble is influenced by the epigenetic state constraints put on the region by its epigenetic state. These constraints may take the form of domain boundaries, where different parts of the chromosome regions are constrained in self-interacting patterns, or by associations with nuclear structures such as the nucleolus. Overall, constraints on the chromosome structure and boundaries between different types of regions

decrease structural ensemble SSI and result in more discrete structures while regions with more uniform linear chromatin states exhibit a more continuous structural landscape with higher SSI.

It might seem likely that the active or inactive chromatin status of a region would be the primary contribution of epigenetics to the structural fluctuation of a region. That is, one might expect an inactive region to have high structural ensemble SSI and an active region low ensemble SSI or vice versa. However, we find that this is not the case. Instead, the uniformity of the epigenetic state appears to be a key factor, such that a uniformly active region shares structural landscape properties with a uniformly inactive region (both have high SSI) while a region divided between alternating states behaves differently (low SSI). This leads us to the idea that alternating epigenetic states place constraints on the possible conformations and thus create a more disjointed, rather than continuous, structural ensemble.

The idea that epigenetic state-mediated constraints are key to shaping dynamic structural landscapes is echoed again by the results we observe in loop extrusion simulations and cohesin depletion data. Previous work has shown that compartmentalization and loop extrusion can act in opposition to each other [69]. When cohesin is removed, we see that a region consisting of multiple epigenetic states can form subdomains and is thus more likely to adopt more discrete structures, resulting in a decrease in ensemble SSI compared to the state where cohesin is active. When cohesin is active, there is a chance of the entire region being pulled into the same loop, creating a more continuous landscape of possible structures (higher SSI). In contrast, when a region exists in a uniform epigenetic state, we see that the removal of cohesin does not result in a reduction of SSI. As before, cohesin removal is expected to reduce the active movement of chromatin, but in this case, the region is entirely in the same compartment, so there are no additional constraints that would lead to the formation of discrete states in the cohesin-depleted structural landscape. Our observations are consistent with the previously published idea that cohesin loop extrusion acts to increase chromosome conformation dynamics, promoting distal contacts and countering the tendency of domains to settle into more discrete sub-structures [69,70]. Our loop extrusion simulations suggest that in a uniform epigenetic state, the region with more boundaries would adopt more discrete structures and thus have less landscape continuity. But, loop extrusion alone, without consideration of interactions between similar chromatin states, cannot reproduce the differences in structure variability we see between different cell types.

The analysis of cohesin depletion results also reiterate the value of considering both the structural landscape and the ensemble SSI of that landscape. The structural landscape of a region can change (adopting a different range of radius of gyration or domain structures) while the continuity (SSI) stays the same. Alternately, the ensemble SSI can change while the structural landscape is overall conserved, indicating cases where the range and types of structures a region can adopt are the same, but the region is more often in certain discrete structures within that range rather than continuously sampling the range.

## Supporting information

**S1 Table. ENCODE accession IDs for the eleven epigenetic marks from three different cell-types.**
(XLSX)

**S2 Table. Parameters values for Lennard-Jones Potential $U_P(r)$ used in the polymer model.**
Each entry represents the interaction strength and distance cutoff (separated by |) between beads of that two specific types. Energy values and distance cutoff values are represented in

terms of $k_B T$ and $\sigma$ respectively.
(XLSX)

**S1 Fig.** Assignment of different chromatin states based on relative enrichment of different histone marks using ChromHMM tool for chr21:28–30 Mb genomic region from three different cell-types—IMR90, K562 and A549 at 30 Kb resolution.
(PNG)

**S2 Fig.** a) Pearson's correlation between chromosome structural landscape PC1 values and radius of gyration ($R_g$) of single-cell structures (calculated from 3D position traces) for IMR90 chr21:28–30 Mb genomic region. b) HiCRatio score of PC2 ordered structures belonging to different PC1 groups for IMR90 chr21:28–30 Mb genomic region, calculated by HiCRatio approach (with a 300 kb window size). Here, PC1G1 refers to the group1 (G1; smallest $R_g$ structures) based on PC1. Within PC1G1, then PC2G1 means the group1 (G1) based on PC2. Successive higher numbered PC1 groups have higher $R_g$ structures. c) For K562 chr21:28–30 Mb genomic region, structures are first divided into three groups based on the PC1 ordering. Within each group, structures are further divided into three subgroups based on their PC2 ordering. To display the domain reorganization along PC2 in K562, averaged distance maps from only groups 2 and 3 from PC1 are shown here. d) For the same groups as in c, HiCRatio (with a 300 kb window size) scores of structures are shown. Colored lines represent HiCRatio scores from PC2 subgroups within two different PC1 subgroups (top = smaller $R_g$ and bottom = larger $R_g$)
(PNG)

**S3 Fig.** For IMR90 chr2 structure data, structures are first divided into ten groups based on the PC1 ordering. Within each group, structures are further divided into ten subgroups based on their PC2 ordering. To display the domain reorganization phenomenon along PC2 in IMR90 chr2, only PC1 subgroups 6 (lower $R_g$) and 10 (higher $R_g$) are shown here. Heatmaps display the average distances for structures within each PC2 subgroup in each PC1 subgroup.
(PNG)

**S4 Fig.** For IMR90 chr21 entire q-arm structure data, structures are first divided into sixteen groups based on the PC1 ordering. Within each group, structures are further divided into sixteen subgroups based on their PC2 ordering. To display the domain reorganization phenomenon along PC2 in IMR90 chr21, only PC1 subgroups 12 (lower $R_g$) and 16 (higher $R_g$) are shown here. Heatmaps display the average distances for structures within each PC2 subgroup in each PC1 subgroup.
(PNG)

**S5 Fig.** a,c) Hierarchical clustering of IMR90 (a) and K562 (c) structures for chr21:28–29.5 Mb genomic segment, based on $Q$ similarity metric as described in Cheng *et al.* (1). To assign structures to the different clusters, the trees are cut at 0.7. b,d) Proportion of structures in different clusters obtained from IMR90 (b) and K562 (d) structural ensembles.
(PNG)

**S6 Fig.** a) Chromosome structural landscape of IMR90 and K562 chr21:28–30 Mb chromatin regions constructed for different numbers of cells. Points are colored according to the radius of gyration (Rg) of the corresponding structures (dark blue–smaller Rg and dark red–larger Rg). b) Chromosome structural ensemble SSI distributions for chr21:28–30 Mb segment from IMR90 and K562 for different number of cells (shown as n). c) Chromosome structural ensemble SSI distributions for the same chr21:28–30 Mb segment from IMR90 and K562 for a lower (35th percentile) and higher (65th percentile) correlation matrix cutoff compared to the cutoff

 

used in the manuscript (50th percentile). The inset (dotted lines) shows a zoom-in to the comparison between IMR90 (left) and K562 (right) at the 65th percentile cutoff. Here, for resampling 200 single-cell structures are selected randomly each time.
(PNG)

**S7 Fig.** a) Chromosome structural landscape of IMR90 and K562 chr21:28–30 Mb chromatin regions constructed for different numbers of cells after removing extremely compact or large conformations. Points are colored according to the radius of gyration (Rg) of the corresponding structures (dark blue–smaller Rg and dark red–larger Rg). b) Chromosome structural ensemble SSI distributions for chr21:28–30 Mb segment from IMR90 and K562 for different numbers of cells, before (green) or after (blue) removing extremely compact or large conformations ("outliers"). 200 single-cell structures are selected randomly each time for resampling.
(PNG)

**S8 Fig.** a) Population-averaged distance maps of active and inactive chrX from IMR90 cell at TAD resolution. b) Jointly constructed chromosome structural landscape of active and inactive chrX from IMR90 cell. c) Bin Variability (BSI standard deviation) of active and inactive chrX, where a higher value represents a higher level of structure variation for that bin across the population and vice versa. d) Chromosome structural ensemble SSI distributions for chr2 segments of varying length. Here, for resampling 300 single-cell structures are selected randomly each time.
(PNG)

**S9 Fig.** a) Representative polymer models of IMR90 and K562 for chr21:28–29.5 Mb genomic segment, obtained from different time points of the simulation. The color of the beads shows corresponding epigenetic states. b) Chromatin states of six different 2 Mb spanning genomic regions of IMR90 chr21 at 50 kb resolution. The chromatin states are inferred based on 11 different histone marks for each cell-type by ChromHMM tool. c) Epigenetic state similarity of different 2 Mb spanning genomic regions of IMR90 chr21 q-arm with the K562 28–30 Mb region using binary comparison. Here, a higher comparison value shows higher similarity with the K562 epigenetic state. d) Chromosome structural ensemble SSI distributions for 31 different 2 Mb spanning genomic regions of IMR90 chr21. For resampling, 300 single-cell structures are selected randomly each time. Mean boundary strength of those regions inferred by HiCRatio have been shown in green color. e) Pearson's correlation between the mean boundary strength and mean ensemble SSI of the 31 different genomic regions obtained from IMR90 chr21.
(PNG)

**S10 Fig.** a) Population-averaged distance maps of IMR90 and K562 chr21:28–30 Mb genomic region from loop extrusion polymer simulations (see Methods) at 1 kb resolution. b) Chromosome structural ensemble SSI distributions for chr21:28–30 Mb from loop extrusion simulated IMR90 and K562. For resampling, 500 single-cell structures are selected randomly each time from a single run (left) or across multiple runs (right). c,d) Jointly constructed chromosome structural landscapes of chr21:28–30 Mb (c) or chr21:34–37 Mb (d) of HCT116 for WT and +Auxin conditions. Points are colored according to the radius of gyration (Rg) of the corresponding structures (dark blue–smaller Rg and dark red–larger Rg). e) Bin Variability of chr21:28–30 Mb genomic region for wild-type (WT) and cohesin-depleted (+Auxin) conditions. f) Bin Variability of chr21:34–37 Mb genomic region for wild-type (WT) and cohesin-depleted (+Auxin) conditions.
(PNG)

## Acknowledgments

The authors would like to thank Dr. Rebeca San Martin for valuable suggestions and insightful discussions in the development of the research project. We thank Dr. Tian Hong for sharing of computational GPU resources.

## Author Contributions

**Conceptualization:** Priyojit Das, Tongye Shen, Rachel Patton McCord.

**Data curation:** Priyojit Das.

**Formal analysis:** Priyojit Das.

**Funding acquisition:** Rachel Patton McCord.

**Investigation:** Priyojit Das.

**Methodology:** Priyojit Das, Tongye Shen.

**Project administration:** Rachel Patton McCord.

**Software:** Priyojit Das.

**Supervision:** Rachel Patton McCord.

**Writing – original draft:** Priyojit Das, Rachel Patton McCord.

**Writing – review & editing:** Priyojit Das, Tongye Shen, Rachel Patton McCord.

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
