## [Decision Letter · Decision Letter 0]

7 Jun 2022

Dear Dr. McCord,

Thank you very much for submitting your manuscript "Characterizing the variation in chromosome structure ensembles in the context of the nuclear microenvironment" for consideration at PLOS Computational Biology. As with all papers reviewed by the journal, your manuscript was reviewed by members of the editorial board and by several independent reviewers. The reviewers appreciated the attention to an important topic. Based on the reviews, we are likely to accept this manuscript for publication, providing that you modify the manuscript according to the review recommendations.

As you will see, Reviewer 2 has made a number of comments and questions designed to improve the readability of the manuscript. We would encourage you to carefully take these into account and modify the manuscript accordingly.

Sincerely,

Carl Herrmann, Ph.D.

Associate Editor

PLOS Computational Biology

Sushmita Roy

Deputy Editor

PLOS Computational Biology

[LINK]

Reviewer's Responses to Questions

**Comments to the Authors:**

Reviewer #1: In this paper, the authors designed a statistical analysis technique to analyze single-cell chromosome imaging data. I found the results interesting and the analysis technically sound, so I support its publication as is.

Reviewer #2: Identifying and characterizing variability within a large set of genomic structures and understanding its origins and mechanisms is a long-standing and important challenge for genome organization. Das et al. develop a method to approach this question quantitatively. The authors develop a pipeline for analyzing ensembles of single-cell chromosome structure data. The purpose is to characterize the hetero- or homogeneity of the structural ensemble, characterize the variability of distances between different individual genomic loci, and begin to identify the origins of the variability within the ensemble. They analyze data from recent imaging experiments, which mapped distances between genomic loci along several chromosomes / chromosomal regions. They find that the compaction/size of the chromosome is the main mode of variability, but factors such as epigenetic state and the presence and positions of genomic boundaries lead to genomic structural variability between cells too. Furthermore, the overall distribution of observed structures is altered by factors such as cell type and cell cycle phase.

The manuscript is generally well written, although there are some places where the methodology of the calculation is difficult to understand. The analysis generally supports the conclusions, except for a few points that could be refined to more narrowly state the conclusion. My main questions regard the robustness of the method (across repeat experiments, different genomic regions, and changes in thresholds applied in the analysis), requirements (in terms of number and quality of samples), and the interpretation of the results (in terms of the meaning of the differences in the “structural similarity index”, etc.). Another issue is how this method might be used to identify new properties of chromosome structure or whether it is better suited for providing quantitative/statistical support for existing hypotheses. These questions should be addressed in the revision. Nonetheless, I think this manuscript is an interesting and useful contribution to the existing literature on genome structure and its analysis, and I would support of acceptance with revisions addressing my comments below.

1. The description of pairwise structure-structure similarity matrix is difficult to understand. After several re-readings of the relevant paragraph, I can see that all of the elements of the calculation are present, but a few extra words to be clear that, for example, each dot product is a single number from the product of two matrices that is then put into another matrix (on which PCC will be calculated, etc.), among other steps in the computation, would be helpful.

2. The main metric, SSI, is difficult to interpret. Because SSI is scaled into arbitrary units, it mainly gives a qualitative comparison between different datasets despite being a calculable quantity. But it is unclear what the size of the effects are. What does a 10% difference in SSI mean for the distribution of single-cell structures? It might be helpful to measure SSI for some model structures such as a (confined) Rouse polymer or a highly crosslinked chain.

3. The authors refer to a pairwise structure-structure “similarity” matrix constructed from dot products of median-centered distance matrices. “Similarity” instead of “correlation” seems slightly misleading to me here. Further, it may be better to rescale the dot products by the product of the mean median-centered distance of each structure (in the same way that correlations of fluctuations in physical systems are typically normalized). This would avoid the following problematic scenario: consider two distance matrices identically equal to the median matrix; they would have a similarity of zero despite being the same (since the median-centered matrices would be identically zero).

4. The analysis could be improved if the authors could be more systematic about the trends of SSI with variables such as number of cohesin or compartment boundaries and variability of epigenetic states. As it stands, a small number of examples are compared, and plausible explanations are given, but it is unclear how well these trends and explanations hold up over larger chromosomal regions, more genomic regions, or even the entire genome.

5. The authors state that higher uniformity of nucleolus association leads to higher continuity of states at the structural ensemble level. But how do we know the direction of causality; could the structure of ensemble-space instead facilitate more consistent nucleolus association? More generally, can the direction of causality be established within this pipeline for phenomena that are not already (somewhat) well understood, such as loop extrusion and compartments/polymer-polymer phase separation?

6. How many single-celled structures are needed to perform this analysis effectively, and for example, reliably compute SSI?

7. How sensitive are calculations and analysis to thresholds used for distance and correlation matrices?

8. In several places, the authors refer to evaluating “dynamics” or contributions (e.g., of epigenetics) to dynamics, but the data is an ensemble of snapshots from different single cells; therefore, dynamics can at best be inferred from the analysis of the structure ensemble.

9. It would be useful for the authors to include a short explanation of the meaning and use of the Shannon-Jayne entropy for context, especially since the main metric studied in the manuscript relies on this entropy.

10. The conclusions about the heterogeneity of regions with less uniform epigenetic marks are interesting. Can the authors comment on whether this is in line with expectations from polymer physics for heteropolymers?

11. The low resolution of the figures (particularly figure 5) and the small label sizes on the axes of the matrices in Figs. 1b,c make the figures difficult to read.

12. Fig 1b panel 3 says mean-centered instead of median-centered.

**Have the authors made all data and (if applicable) computational code underlying the findings in their manuscript fully available?**

Reviewer #1: Yes

Reviewer #2: Yes

PLOS authors have the option to publish the peer review history of their article (what does this mean?). If published, this will include your full peer review and any attached files.

Reviewer #1: No

Reviewer #2: No

Figure Files:

Data Requirements:

Reproducibility:

References:

---

## [Editor Report · Decision Letter 1]

15 Jul 2022

Dear Dr. McCord,

We are pleased to inform you that your manuscript 'Characterizing the variation in chromosome structure ensembles in the context of the nuclear microenvironment' has been provisionally accepted for publication in PLOS Computational Biology.

Best regards,

Carl Herrmann, Ph.D.

Associate Editor

PLOS Computational Biology

Sushmita Roy

Deputy Editor

PLOS Computational Biology

---

## [Editor Report · Acceptance letter]

10 Aug 2022

PCOMPBIOL-D-22-00610R1 

Characterizing the variation in chromosome structure ensembles in the context of the nuclear microenvironment

Dear Dr McCord,

I am pleased to inform you that your manuscript has been formally accepted for publication in PLOS Computational Biology. Your manuscript is now with our production department and you will be notified of the publication date in due course.

With kind regards,

Zsofia Freund
